# Vinblastine/Methotrexate for Debilitating and Progressive Plexiform Neurofibroma in Children and Young Adults with Neurofibromatosis Type 1: A Phase 2 Study

**DOI:** 10.3390/cancers15092621

**Published:** 2023-05-05

**Authors:** Chelsea Kotch, Kristina Wagner, J. Harris Broad, Eva Dombi, Jane E. Minturn, Peter Phillips, Katherine Smith, Yimei Li, Ian N. Jacobs, Lisa M. Elden, Michael J. Fisher, Jean Belasco

**Affiliations:** 1Division of Oncology, Department of Pediatrics, Children’s Hospital of Philadelphia, Philadelphia, PA 19104, USA; 2Perelman School of Medicine, University of Pennsylvania, Philadelphia, PA 19104, USA; 3Department of Anesthesiology, Valley Medical Center, Renton, WA 98055, USA; 4Pediatric Oncology Branch, Center for Cancer Research, National Cancer Institute, Bethesda, MD 20892, USA; 5Department of Biostatistics, University of Pennsylvania, Philadelphia, PA 19104, USA; 6Division of Otolaryngology, Department of Pediatrics, Children’s Hospital of Philadelphia, Philadelphia, PA 19104, USA

**Keywords:** plexiform neurofibroma, neurofibromatosis type 1, chemotherapy, clinical trial

## Abstract

**Simple Summary:**

Limited effective therapies exist for the treatment of neurofibromatosis type 1 (NF1)-associated plexiform neurofibroma (PN). The aim of our prospective clinical trial was to evaluate the activity and safety of vinblastine and methotrexate (conventional cytotoxic chemotherapies) given on a metronomic schedule in children and young adults with NF1 and PN. Of the 23 participants evaluable for treatment outcomes and toxicities, 14 completed all protocol therapy. There were no participants that discontinued therapy due to dose-limiting toxicities. In addition, there were no participants that demonstrated a partial response. VBL/MTX was well-tolerated but did not result in objective volumetric responses.

**Abstract:**

Limited therapies exist for neurofibromatosis type 1 (NF1)-associated plexiform neurofibroma (PN). For this reason, the activity of vinblastine (VBL) and methotrexate (MTX) was evaluated in children and young adults with NF1 and PN. Patients ≤ 25 years of age with progressive and/or inoperable NF1-PN received VBL 6 mg/m^2^ and MTX 30 mg/m^2^ weekly for 26 weeks, followed by every 2 weeks for 26 weeks. Objective response rate was the primary endpoint. Of 25 participants enrolled, 23 were evaluable. The median age of participants was 6.6 years (range 0.3–20.7). The most frequent toxicities were neutropenia and elevation of transaminases. On two-dimensional (2D) imaging, 20 participants (87%) had stable tumor, with a median time to progression of 41.5 months (95% confidence interval 16.9, 64.9). Two of eight participants (25%) with airway involvement demonstrated functional improvements including decreased positive pressure requirements and apnea-hypopnea index. A post hoc three-dimensional (3D) analysis of PN volumes was completed on 15 participants with amenable imaging; 7 participants (46%) had progressive disease on or by the end of therapy. VBL/MTX was well-tolerated but did not result in objective volumetric response. Furthermore, 3D volumetric analysis highlighted the lack of sensitivity of 2D imaging for PN response evaluation.

## 1. Introduction

Neurofibromatosis type 1 (NF1) is an autosomal dominant cancer predisposition syndrome occurring in approximately 1 in 3000 people worldwide. NF1 occurs due to an inherited or sporadic mutation in the *NF1* tumor suppressor gene [1]. Patients with NF1 are at increased risk for the development of both central and peripheral nervous system tumors, including plexiform neurofibromas (PN) [2,3]. PN are benign peripheral nerve sheath tumors that occur in up to half of patients with NF1. PN may be asymptomatic or can cause significant morbidity such as pain, disfigurement, neurologic dysfunction such as weakness, or impinge on vital structures such as the airway [4,5]. In addition, PN may become life-threatening when significant impact on vital structures or malignant transformation occurs [1,6,7].

Effective treatment options for PN are limited [8,9,10,11,12]. Historically, treatment was limited to surgical resection. However, due to the infiltrative nature of these tumors, PN are often not amenable to resection, and surgical intervention can lead to worsening pain and neurologic deficits [6,13,14,15]. Numerous prior studies of targeted agents for PN have failed to prolong progression-free survival or achieve objective (partial) responses [8,9,10,11,12]. However, recently, mitogen-activated kinase (MEK) inhibitors have shown impressive efficacy in clinical trials for the treatment of PN with high partial response rates [4,16], resulting in the Food and Drug Administration (FDA) approval of selumetinib in 2020 for the treatment of pediatric patients with PN. However, up to one third of participants treated with selumetinib fail to demonstrate a meaningful response, and a small subset are unable to tolerate therapy due to the associated toxicities [4]. At present, selumetinib remains the only FDA approved agent for the treatment of PN, yet the label is limited to children aged 2 years and older with symptomatic, inoperable PN. Thus, the identification of additional medical therapies for PN remains a priority.

Conventional cytotoxic chemotherapy has historically had little activity for PN. However, in desmoid-type fibromatosis, a disease historically considered to have clinical and histologic similarities to PN, vinblastine (VBL), and methotrexate (MTX) resulted in high objective response rates and were well tolerated [17]. Furthermore, these agents are not considered to have malignant potential, which is of important consideration in selection of therapies for patients with NF1 [18]. Thus, we evaluated VBL/MTX in a phase 2 clinical trial for patients with NF1 and symptomatic or progressive, inoperable PN. 

## 2. Materials and Methods

### 2.1. Study Design and Population

The study was approved by the institutional review board at the Children’s Hospital of Philadelphia. Written informed consent was obtained from patients aged 18 years or greater or from parents/guardians of children <18 years. The trial was registered with www.clinicaltrials.gov, identification number NCT00030264. Patients were enrolled at the Children’s Hospital of Philadelphia between 1 March 2001 and 30 May 2012. Inclusion criteria included ages ≤ 25 years with a diagnosis of NF1, as defined by the NIH Consensus Conference criteria [19], an unresectable PN either with significant morbidity or progression in the year prior to enrollment (defined as a measurable increase in the sum of the product of the two longest perpendicular diameters of the index lesion over <12 months prior to enrollment), and adequate bone marrow and hepatic function. Exclusion criteria included Lansky or Karnofsky performance statuses of less than 60, pregnancy, or exposure to an investigational agent or chemotherapy within 30 days of enrollment. The primary objectives were to determine objective response rate and time to progression. 

### 2.2. Therapy

Participants received VBL 6 milligrams (mg) per meter squared (m^2^) per dose (maximum dose 10 mg) intravenous (IV) weekly for 26 weeks and then every 2 weeks for 26 weeks or until disease progression. MTX dosing was 30 mg/m^2^/dose IV weekly for 26 weeks and then every 2 weeks for 26 weeks or until disease progression. Participants could receive a maximum of 52 weeks of therapy. Participants were considered evaluable if they received at least one dose of protocol therapy. Participants were removed from therapy for progressive disease, intolerable toxicity, patient/family request, or noncompliance. Dose-limiting toxicities (DLT; hematologic or non-hematologic) were defined as grade 3 or greater toxicities. If the toxicity resolved to Grade 2 or less within 14 days, therapy was restarted at a 25% lower dose. Toxicities requiring removal from protocol therapy included any therapy related DLT that recurred after a second dose reduction (>50% from initial dose) and any grade 4 non-hematologic toxicity. 

### 2.3. Study Evaluations

Magnetic resonance imaging (MRI) of the target PN was required within 2 weeks of the first dose of therapy. Imaging was obtained every 3 months while in therapy. Patients were considered to have airway involvement of PN if enrollment imaging demonstrated any degree of mechanical compression at any location from the oropharynx to major bronchi. For patients with tumors affecting the airway, video laryngoscopy, polysomnography, and pulmonary function tests (if age appropriate) were obtained at time of enrollment and end of therapy at a minimum.

### 2.4. Safety Monitoring

Monitoring included physical examination and laboratory evaluations (including complete blood counts prior to each dose and monthly comprehensive metabolic panel). Adverse events (AE) were graded according to the National Cancer Institute Common Terminology Criteria for Adverse Events version 4.0. Participants were considered evaluable for toxicity if they received at least one dose of protocol chemotherapy. Toxicities were summarized by frequencies of each toxicity overall. For each toxicity, the worst grade overall for each patient was calculated and summarized.

### 2.5. Outcomes Measures

The primary outcome of the study was objective response rate. An additional primary outcome was time to progression (TTP), which was defined by imaging as the first two-dimensional (2D) radiologic progression or by functional decline from the start of protocol therapy. Functional decline was defined as deterioration of functional status attributed to the target PN, such as decline in neurologic examination (e.g., weakness) over time and/or worsening of objective measures on functional testing such as polysomnography or pulmonary function testing. 2D imaging response was calculated using the sum of the product of the two longest perpendicular diameters of the index lesion. Partial response (PR) was defined as a decrease in tumor size by ≥25%. Progressive disease (PD) was defined as an increase in tumor size of ≥25%. Stable disease (SD) was defined as <25% to >25% change in tumor size. A post hoc three-dimensional (3D) volumetric MRI response evaluation was performed at the National Cancer Institute for the target PN as previously described [20,21], with evaluation of enrollment and final on-study MRI scan. PR was defined as a decrease in the volume of the target lesion by ≥20% compared with baseline. PD was ≥20% increase in tumor volume at any time on therapy compared to baseline. SD was defined as any change in tumor volume that did not quality as a PR or PD [21].

Analysis of functional responses for the cohort of participants with airway involvement were guided by the Response Evaluation in Neurofibromatosis and Schwannomatosis (REiNS) sleep and pulmonary outcomes recommendations [22]. The primary sleep endpoint evaluated on polysomnography was the apnea–hypopnea index. A clinically meaningful improvement was defined as an absolute decrease in AHI by ≥5 events per hour. Subjective improvements were described. 

### 2.6. Statistical Analysis

The primary trial objective was to determine whether the use of VBL/MTX in children and young adults with NF1 and progressive and/or inoperable PN resulted in objective responses. Proportions of response categories (PR, SD, PD) were calculated for both 2D and 3D imaging outcomes. For assessment of TTP, a Kaplan–Meier survival curve was calculated for 2D imaging and/or functional progression and plotted with a 95% confidence interval around the median. All analyses were performed using Stata 15 (StataCorp, College Station, TX, USA).

## 3. Results

### 3.1. Patient Characteristics

Twenty-five patients were enrolled, but two were unevaluable as they did not receive any protocol therapy after initial enrollment. Thus, 23 participants were evaluable for toxicities and tumor responses. Of these, 15 participants (65%) enrolled with radiographically progressive PN on 2D imaging and 8 (35%) enrolled with PN considered to be causing significant morbidity or functional impairment (of these 4 had stable tumor prior to treatment and 4 did not have pre-study imaging available for assessment) (Table 1). The median age at enrollment was 6.6 years (range 0.3–20.7 years). The most frequent target PN location was head and neck (44%), followed by the trunk (35%) (Table 1). Of the 14 tumors with any involvement of the neck, 9 (64%) had airway involvement. Five (56%) of the 9 participants with airway involvement had associated functional impairments at enrollment. Fourteen (61%) of the 23 participants completed protocol therapy. Of the 9 (39%) that did not complete protocol therapy, 3 (13%) progressed on therapy (weeks 13, 14, and 40), 1 withdrew due to progression of a low-grade glioma (week 14), and 5 withdrew from therapy due to patient/caregiver choice (weeks 1, 12, 14, 17, 36); only 1 participant withdrew for low-grade intolerable adverse events (grade 1 neuropathy). One of the participants who progressed on therapy (week 13), subsequently, died from airway complications related to their target PN. One subject died from a malignant peripheral nerve sheath tumor 18 months after completion of protocol therapy, despite the stable disease of the target PN. 

### 3.2. Clinical Safety and Tolerability 

VBL/MTX was well tolerated over multiple cycles. No participants discontinued therapy due to DLTs, and there were no grade 5 AEs. The most common AEs of any grade were hematologic and gastrointestinal (Table 2). Eleven participants (48%) required at least one dose reduction of VBL and/or MTX while on study; 8 participants required dose reductions of VBL and 6 required dose reductions of MTX. Indications for reduction of VBL included neutropenia, neuropathy, constipation, and anorexia. Indications for MTX dose reductions included neutropenia, elevation of transaminases, nausea/vomiting, and anorexia. 

### 3.3. Radiographic Tumor Response 

Of the 23 evaluable participants, at final follow up on protocol therapy, there were no partial responses observed. Twenty participants (87%) had SD on 2D imaging, and three participants (13%) had PD. All three participants who progressed on therapy had progressive tumor on imaging at study entry. The median time to 2D and/or functional progression was 41.5 months (95% CI 16.9, 64.9). A post hoc analysis of three-dimensional, volumetric imaging response was performed; 15 participants (65%) had imaging amenable to volumetric MRI analysis. The primary reasons for lack of evaluable volumetric imaging included incomplete/inadequate tumor coverage, fat interference, and lack of short tau inversion recovery sequences. Of these 15, 8 participants (53%) had SD, and 7 (47%) had PD on or by the end of therapy (Table 3). There was no significant difference in the volumetric response by tumor location (*p* = 0.725). Of note, the two participants noted to have PD by 2D measurements were also identified as PD by 3D analysis. However, 5 of the 13 identified as SD by 2D measurements were actually PD by 3D analysis (Appendix A). 

### 3.4. Functional and Primary Sleep Outcomes of Airway Cohort

Nine participants had baseline involvement of airway by PN; eight of these participants had serial functional evaluations throughout protocol therapy and were considered evaluable for functional and sleep outcomes, and six had imaging amenable to volumetric analysis, of which 50% had PD on therapy (Table 4). Only one subject had a baseline AHI of more than five, which is considered the lower limit needed to see a meaningful effect of treatment on airway function [22]; this subject had a reduction in AHI from 5.2 to 1.4, but did not have imaging amenable to volumetric analysis. Another subject had a reduction in continuous positive airway pressure requirements from 10 cm of water pressure (cm H_2_O) to 4 cm H2O, despite SD on volumetric analysis. Subjective improvements were also noted, with one subject endorsing improvement in symptoms such as vocal quality and stridor. Three participants demonstrated a decline in function (including increased AHI on therapy). Overall, the proportion of participants free from functional decline while on protocol therapy was 63%. Of note, impulse oscillometry was not routinely assessed clinically for the participants and was not required as part of the trial. 

## 4. Discussion

This clinical trial represented the first analysis of the activity of VBL/MTX for patients with NF1 and progressive and/or inoperable PN. While initial 2D imaging analysis suggested VBL/MTX delayed the time to tumor progression in pediatric and young adult participants with NF1 and PN, a post hoc 3D analysis determined that VBL/MTX did not result in any PRs, with 47% of those evaluable for volumetric analysis demonstrating PD while on therapy. Thus, VBL/MTX did not demonstrate activity as defined by the primary objectives of our study. 

The discrepancy of imaging outcomes determined between the 2D and 3D imaging techniques further supported recommendations for the use of 3D volumetric imaging in the assessment of PN response to treatment [8,21,23]. Of the participants with imaging amenable to volumetric analysis, 33% with PD on the post hoc 3D analysis was misclassified as SD on 2D outcomes assessment. Linear (2D) measurements can be highly variable and less sensitive to tumor growth in non-spherical tumors, such as PN, thereby making accurate assessments of progression challenging. Volumetric (3D) analysis accounts for all parts of the PN and reflects the actual size of the tumor more closely than linear measurements, with enhanced detection of small changes over time [21]. Thus, the difference in rate of PD by 2D and 3D measurements represents improved accuracy of measurements of tumor growth. Prospective use of 3D imaging for this study may have shortened the duration of exposure to immunosuppressive chemotherapy based on earlier recognition of progression, thereby decreasing the duration of risk of adverse effects of therapy [24,25]. As the utilization of medical therapies such as selumetinib for PN increases, clinicians should be aware of the relative limitations of interpreting response or progression with 2D imaging. Given the time and resource burden of volumetric MRI analysis, the results of this study, again, highlighted the need for the development of methods that could be easily performed across institutions and incorporated into routine clinical practice. 

Despite the apparent lack of activity of VBL/MTX for PN, of interest were the two patients with airway tumors and functional improvements on therapy. Despite a lack of imaging response, these two participants demonstrated a measurable improvement in primary sleep endpoint measures, with a reduction in AHI and a decline in CPAP requirement. This discordance between imaging and function was not entirely unexpected, as resistance to airflow is inversely proportional to the radius of the airway (Poiseuille’s law); thus, a minimal reduction in a focal area of tumor may have a clinically meaningful impact on airflow. [26] However, a lack of robust natural history data of functional outcomes in PN makes interpretation of these results challenging. It is unknown if the improvements observed or the proportion of participants free from functional decline while on protocol therapy (63%) differs significantly from the expected natural history. Furthermore, it is unclear if these functional improvements may be observed spontaneously in the setting of patient growth and development (e.g., increase in airway diameter with age in young children) or represent a true functional response to therapy. In addition, changes in AHI in the pediatric population may also relate to change in tonsil/adenoid tissue that peaks in size between 5 to 7 years of age and then atrophies over time [27,28], thereby impacting the interpretation of primary sleep outcome measures in children with NF1 and PN. The present study highlighted the need for additional objective natural history data regarding functional outcomes for patients with PN and airway involvement. Such data will be required in order to interpret clinical trial results more accurately. 

In review of the existing literature, there is a paucity of objective functional outcomes described for PN with airway involvement. In addition, limited clinical trials of PN have incorporated the recommended functional and sleep outcomes proposed by REiNS, further hindering comparisons of objective outcomes between studies [22]. In a retrospective study examining the association of volumetric changes in PN and development of clinical morbidities in NF1, only 2 of 41 participants had airway morbidity; objective changes in primary sleep endpoints such as AHI or CPAP requirements were not quantified for the period of observation reported [29]. In addition, in the pilot phase 2 study of imatinib, subjective improvements in disease symptoms were reported, including improved dyspnea and resolution of snoring/disruptive sleep pattern, yet no objective measures were described [30]. The largest cohort of patients with airway outcomes was described in the phase 2 study of selumetinib for PN, where 16 participants had airway impairment at baseline. Ten of these participants had polysomnography data, and there were no meaningful changes in airway obstruction during sleep [4]. Importantly, none of the patients enrolled in the selumetinib study had a baseline AHI of more than 5, thereby limiting the ability to evaluate patients based on AHI for a meaningful treatment effect. However, five participants (45%) did show a clinically meaningful improvement in airway resistance measured by impulse oscillometry [4,22]. Collectively, the present and prior studies highlighted the need for uniform assessments and reporting of functional and sleep outcomes to allow for more informed determinations of the efficacy of experimental therapeutics for PN with airway impairment. 

Additional limitations of the present trial include a small sample size, a lack of consistent secondary functional measures for the airway cohort, and the absence of serial functional evaluations for participants with tumors not affecting the airway. Given that only one subject with airway impairment had an evaluable baseline AHI, this study would have derived benefit from the inclusion of secondary measures of pulmonary function such as impulse oscillometry; however, REiNS recommendations were not yet published at the time of study design, and impulse oscillometry was not routinely captured as part of clinical care or study evaluations. In addition, progression status prior to enrollment was not able to be determined using volumetric analysis; thus, the rates of tumor stabilization could not be evaluated. Additionally, as the study did not require off study MRI evaluations, there was insufficient quality and quantity of imaging to calculate a meaningful volumetric time to progression; however, given that 46% of patients with volumetric analysis demonstrated PD on study, it was highly unlikely MTX/VBL resulted in the significant prolongation of TTP compared to the historical clinical trial, placebo arm, or control cohorts [8]. Finally, the understanding of activity of VBL/MTX for PN in adults was limited, as this study restricted enrollment to participants less than 25 years of age. 

## 5. Conclusions

Overall, although VBL/MTX was generally well-tolerated, it did not demonstrate activity for NF1-associated PN. At present, given the recent successes of mitogen-activated protein kinase inhibitors, as well as emerging data with other targeted therapies, we believe that VBL/MTX does not warrant further evaluation in children with NF1 and progressive and/or inoperable PN. In addition, this study in combination with the existing literature highlights the need for more granular, uniform reporting practices of functional outcomes of participants with airway tumors to allow for increased understanding of the efficacy and impact of novel therapies for PN.

## Figures and Tables

**Table 1 cancers-15-02621-t001:** Clinical characteristics of 23 evaluable participants with neurofibromatosis type 1 and plexiform neurofibroma.

Clinical Characteristics	Evaluable Subjects (N = 23)
Median Age at Enrollment, Years (Range)	6.6 (0.3–20.7)
Sex Male Female	10 (43%) 13 (57%)
Target PN Location Head/Neck Neck/Trunk Trunk Only Trunk and Extremity	10 (44%) 4 (17%) 8 (35%) 1 (4%)
Head/Neck PN with Airway Involvement Yes No	9 (64%) 5 (36%)
Indication for Enrollment Radiographic Progression * Functional Decline or Morbidity	15 (65%) 8 (35%)
Prior Treatment of Target PN ** Yes No Unknown	11 (48%) 3 (13%) 9 (39%)

Abbreviations: N, number; PN, plexiform neurofibroma. * Two-dimensional radiographic progression; ** Medical or surgical (no patients received prior treatment with a mitogen activated kinase inhibitor).

**Table 2 cancers-15-02621-t002:** Number and percentage of participants with grade 3 or greater toxicities (worst toxicity grade per patient) during all treatment cycles administered of vinblastine/methotrexate.

Toxicity (CTCAEv4.0)	Grade 3	Grade 4
Hematologic		
Neutropenia	2 (9%)	4 (17%)
Febrile neutropenia	1 (4%)	
Anemia	1 (4%)	
Gastrointestinal		
Anorexia	1 (4%)	
Nausea	2 (9%)	
AST increase	2 (9%)	
ALT increase	2 (9%)	
Infection		
Cellulitis	1 (4%)	
Tracheitis	1 (4%)	
Mucosal infection	1 (4%)	
Fever	3 (13%)	

Abbreviations: CTCAEv4.0, Common Terminology Criteria for Adverse Events version 4.0; AST, aspartate transaminase; ALT, alanine transaminase.

**Table 3 cancers-15-02621-t003:** Clinical characteristics and tumor response of cohort by two-dimensional versus three-dimensional imaging analysis.

Clinical Characteristic and Tumor Response	Two-Dimensional Cohort (N = 23)	Three-Dimensional Cohort (N = 15) *
Median Age at Enrollment, Years (Range)	6.6 (0.3–20.7)	6.4 (1.7–20.7)
Target PN Location Head/Neck Neck/Trunk Trunk Only Trunk and Extremity	10 (44%) 4 (17%) 8 (35%) 1 (4%)	8 (53%) 3 (20%) 3 (20%) 1 (7%)
PN with Airway Involvement	9 (64%)	6 (40%)
Imaging Outcome Partial Response Stable Disease Progressive Disease	0 (0%) 20 (87%) 3 (13%)	0 (0%) 8 (53%) 7 (47%)

Abbreviation: N, number; PN, plexiform neurofibroma. * 15 of 23 subjects had imaging amenable to volumetric magnetic resonance imaging analysis.

**Table 4 cancers-15-02621-t004:** Radiologic and Primary Sleep Endpoints of Cohort with Airway Involvement (N = 8 *).

Subject	Index PN Location	Age at Enrollment	Radiologic Disease Status at Enrollment, 2D	AHI at Enrollment	Radiologic Response at EOT, 2D	AHI at EOT	Radiologic Response at EOT, 3D (% Change)	Additional Functional Changes	Primary Sleep Endpoint (Functional Response)	Time to Functional Progression
1	Neck with severe supraglottic obstruction	15.5 years	PD	NE	SD	NE	SD (9.7)	CPAP decreased from 10 to 4 cm H_2_O	Response	1554 days; CPAP 6-8
2	Neck with compression and deviation of oropharynx	2.3 years	PD	AHI 3.3	SD	AHI 4.9	PD (26.4)	Weaned off PM oxygen, nadir SpO_2_ > 90%	SD	509 days; increased AHI 13.5
3	Neck with mild diffuse tracheal narrowing from upper neck to tracheal carina	4.0 years	SD	AHI 5.2	SD	AHI 1.4	NE	Improved stridor, resolution of cyanotic spells	Response	451 days; increased AHI 2.5 and stridor
4	Neck with severe supraglottic obstruction	4.1 years	PD	AHI 0.4 (tracheostomy)	SD	AHI 0.8	SD (1.6)	N/A	SD	No associated functional decline, decannulated 10 years from enrollment
5	Neck with nasopharyngeal/upper airway deviation	10.3 years	PD	AHI 0.6	SD	AHI 0	NE	N/A	SD	543 days; change on laryngoscopy and swallowing
6	Neck with severe subglottic stenosis	0.3 years	PD	Tracheostomy	PD	Deceased	NE (PD)	N/A	PD	98 days; tracheostomy failure
7	Neck with laryngeal narrowing and mediastinal deviation	3.9 years	SD	AHI 1.8	SD	AHI 25.6 ^%^	PD (53)	N/A	NE ^%^	(Radiographic progression; MEK inhibitor started at 1037 days)
8	Neck with displacement of pharynx and attenuation of airway at C2-3	6.8 years	PD	AHI 1.6	PD	AHI 14.7	PD (27.6)	N/A	PD	273 days; increased AHI 14.7

* One subject did not continue functional evaluations post-enrollment; ^%^ Associated adenoid hypertrophy, AHI returned to baseline after adenoidectomy. Abbreviations: 2D, two-dimensional; AHI, apnea-hypopnea index; CPAP, continuous positive airway pressure; EOT, end of therapy; N, number; N/A, not applicable; NE, not evaluable; PD, progressive disease; PN, plexiform neurofibroma; SD, stable disease.

## Data Availability

The data presented in this study are available on request from the corresponding author. The data are not publicly available due to institutional restrictions.

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
