# Peer review of "Vinblastine/Methotrexate for Debilitating and Progressive Plexiform Neurofibroma in Children and Young Adults with Neurofibromatosis Type 1: A Phase 2 Study"

_cancers, 2023, doi:10.3390/cancers15092621_

Round 1

Reviewer 1 Report

While a "negative study" in terms of outcome, I would consider it critical to publish such results instead of relying on anecdotes about lack of chemotherapeutic efficacy when discussing treatment options and counseling families. Therefore, I commend the authors for putting together this manuscript describing the toxicity and efficacy (or lack thereof) of conventional chemotherapy in NF1 plexiform neurofibromas. The manuscript is well written and concise with a clear explanation of methods and results and acknowledgement of study limitations. It also highlights the importance of volumteric analysis when assessing treatment response in PNs although the feasibility of this outside of major academic sites with adequate experience remains a substantial hurdle preventing widespread utilization. However, there are a few aspects of the manuscript that could be improved with some finer details that can help clarify the methods used -

1) In the methods section - Perhaps consider clarifying what functional progression means or is defined as since this was used as end point for TTP?

2) Nine patients had tumors with airway involvement - please clarify if this was radiographic only or functional airway deficits were noted in all patients.

3) Two patients (among 25) were not evaluable and were excluded from analysis - please clarify the reason?

4) Regarding the post-hoc analysis of 3-dimensional, volumetric imaging response was performed; 15 participants (65%) had imaging amenable to volumetric MRI analysis

- Please clarify reasons imaging were not considered amenable for 3D volumetrics?

- The authors initially state that 15 participants had PD by imaging and 8 had significant morbidity prior to study enrollment. Upon post hoc analysis of 3D volumetrics (if feasible), were you able to ascertain if any patients who had PD by volumetrics analysis prior to enrollment were able to achieve disease stabilization by 3D volumetrics indicating some efficacy of the combination chemotherapy.

Reviewer 2 Report

The authors presented a well-designed, long-term, phase 2 clinical trial on NF1 PN, which showed a lack of ORR after treatment with VBL/MTX low-toxicity chemotherapy. The paper is well-written and the results are very useful to the field of NF1. 

Here are a few minor suggestions and comments: 

Line 74: May the authors cite a reference for the NIH Consensus Conference criteria of unresectable PN? 

Line 178: Only 15 patients had MRI amenable for 3D evaluation. Can the authors discuss why? 

Lines 199-200: A 63% of non-progression rate duing treatment - How does 

Lines 218-223: NF1 PNs usually have complex imaging findings. Were the different PD rate by 2D and 3D measurements mainly due to a much lower threshold for PD used by 3D measurements? i.e. Would the authors consider this difference a "size/definition" issue or more likely a "tumor shape" issue? 
(Mathematically, it seems that if a NF1 PN tumor is a perfect spherical object, the definition of PD as a 25% increase by 2D measurement = ((125%)^0.5)^3 = 39% increase by 3D measurement, vs. the definition of PD as 20% increase by 3D) 
Please discuss. 
Some example case studies/images comparing 2D vs. 3D measurements may be helpful. 

Table 1: May the authors describe more about which and how many lines of prior therapies the 11 patients had had before enrollment to VBL/MTX? The information can be put in a footnote or in the text. Did any one of these patients receive targeted therapy, e.g. a MEK inhibitor? 

Table 2: Although case number is small, I suggest to add in percentage after each number. 

Reviewer 3 Report

Dear authors, 

Below are the suggestions and comments to be include in the manuscript. 

1. Introduction

- Please add the prognosis of this cancer. 

- Explain briefly what are the neurological effects of this cancer. 

- Briefly explain previous study, how is the management of this cancer and what is the outcome. 

-What make this drug better than previous previous standard treatment? 

2. Methodology

- please details out the demographic information about the patients condition

- sample size is small 

- good if author can add more patients with more homogenous  population 

- the patients location of target is varies, good if can group the patient accordingly. 

3. Results

- the outcome is not clear, could you please details it out and explain in detail. 

4. Discussion

- please discuss more on the outcome and how this study improve our management of this type of cancer

Thank you
